# A Fluorescence Sensing Determination of 2,4,6-Trinitrophenol Based on Cationic Water-Soluble Pillar[6]arene Graphene Nanocomposite

**DOI:** 10.3390/s19010091

**Published:** 2018-12-28

**Authors:** Xiaoping Tan, Tingying Zhang, Wenjie Zeng, Shuhua He, Xi Liu, Hexiang Tian, Jianwei Shi, Tuanwu Cao

**Affiliations:** Key Lab of Inorganic Special Functional Materials, Chongqing Municipal Education Commission, School of Chemistry and Chemical Engineering, Yangtze Normal University, Chongqing 408100, China; 18875144840@163.com (T.Z.); 17623010998@163.com (W.Z.); thshuhua@163.com (S.H.); Liuxi201608551226@163.com (X.L.); X15826250409@163.com (H.T.); jianweicn2000@163.com (J.S.)

**Keywords:** cationic pillar[6]arene, host–guest recognition, reduced graphene, trinitrophenol

## Abstract

We describe a selective and sensitive fluorescence platform for the detection of trinitrophenol (TNP) based on competitive host–guest recognition between pyridine-functionalized pillar[6]arene (PCP6) and a probe (acridine orange, AO) that used PCP6-functionalized reduced graphene (PCP6-rGO) as the receptor. TNP is an electron-deficient and negative molecule, which is captured by PCP6 via electrostatic interactions and π–π interactions. Therefore, a selective and sensitive fluorescence probe for TNP detection is developed. It has a low detection limit of 0.0035 μM (S/N = 3) and a wider linear response of 0.01–5.0 and 5.0–125.0 for TNP. The sensing platform is also used to test TNP in two water and soil samples with satisfying results. This suggests that this approach has potential applications for the determination of TNP.

## 1. Introduction

Nitroaromatics (NACs) are toxic and explosive [1]. Nitroaromatic explosives are common components of industrial explosives, such as 2,4,6-trinitrophenol (TNP), 2,4-dinitrotoluene (DNT) and 2,4,6-trinitromethylbenzene (TNT) [2]. TNP is more explosive than TNT, and it has been widely used in the military as well as in objects that include fireworks, dyes and glasses [3]. TNP might harm the skin and eyes and can damage organs. It is an environmental pollutant and harmful to human health. Therefore, rapid, selective and sensitive determination of TNP is increasingly important for security and environmental protection.

TNP is currently determined by using solid-phase microextraction [4], spectrophotometry [5], X-ray imaging [6], gas chromatography (GC) [7], capillary electrophoresis (CE) [8], high-performance liquid chromatography (HPLC) [9], and electrochemical methods [10]. Unfortunately, these methods often need complicated synthetic processes—these are time consuming and complicated. In recent years, fluorescence techniques have been shown to be a promising method for the detection of TNP. They exhibit many advantages over other common detection techniques; they are sensitive, selective and portable [11,12].

For example, Rong et al. [13] developed a label-free fluorescence sensing method that used chemically oxidized and liquid exfoliated g-C_3_N_4_ nanosheets for the determination of TNP with a detection limit of 8.2 × 10^−6^ M. Ma et al. [14] prepared dual-emissive electropolymerization films as fluorescent probe for the detection of TNT. Peng et al. [15] constructed a new boron nitride quantum dots (BNQD)-based turn-off probe for the sensitive detection of TNP based on the strong inner filter effect (IFE) between TNP and the BNQDs—the linear range was 2.5 × 10^−7^–2.0 × 10^−4^ M and the detection limit was 1.4 × 10^−7^ M.

Many fluorescent probes have been prepared for TNP determination including the graphitic carbon nitride (g-C_3_N_4_) nanosheets [14,15], metal-organic frameworks [16], transition-metal dichalcogenide nanostructures [17], graphene quantum dots (GQDs) [18], conjugated polymers (CPs) [19], and other carbon-based nanomaterials [20,21]. Although many fluorescence sensing probes have been reported, the selectivity and sensitivity are relative unsatisfied. Therefore, tools for selective and sensitive detection of TNP are still an unmet need.

Macrocyclic arenes have been widely studied in the field of supramolecular chemistry [22]. After crown ethers, calixarenes, cyclodextrins and cucurbiturils [23], pillar[n]arenes are the fifth class of macrocyclic host molecules, and they are firstly reported by Ogoshi [24]. Pillar[n]arenes mainly consist of pillar[5]arenes and pillar[6]arenes, which are linked by methylene bridges at their para-positions to form a unique rigid pillar architecture. Pillar[n]arenes are important players in supramolecular chemistry because of their easy synthesis, unique pillar shape, symmetrical structure, versatile functionalization, excellent host–guest properties, and natural supramolecular assembly characteristics. They have been widely applied in host–guest chemistry and biomedical material [25]. However, by comparing other conventional macrocyclic hosts, their application in fluorescent probes is rare.

Recently, Shao et al. [26] reported a novel fluorescent supramolecular cross-linked polymer network that could detect TNP in solutions and films via pillar[5]arene-based host-guest recognition. The combination of supramolecular chemistry and conjugated polymer science as well as the co-constructed supramolecular network system can pave the way for new multi-functional fluorescent materials. The using of a water-soluble macrocyclic host that interacts with graphene by π–π stacking can improve selectivity and sensitivity [27,28]. Graphene has unique thermal, electronic and mechanical properties, as well as a high surface area, low cost and low toxicity due to its strictly 2D structure [29]. It has been applied in sensing, drug carriers and other technological fields.

While host-guest recognition has potential applications in many areas, including probe, gene and drug delivery, nanoelectronics, supramolecular polymers, high recognition and binding strength. The guest TNP is an electron-deficient molecule and is easily captured by the electron-rich cavity of the cationic pillar[6]arene. Therefore, we describe a competitive fluorescence sensing platform based on pyridine-functionalized pillar[6]arene and reduced graphene PCP6-functionalized reduced graphene (PCP6-rGO) nanocomposite for TNP determination. The PCP6 is grafted on the surface of reduced graphene (rGO) via π–π stacking to obtain the receptor. The indicator/dye molecule AO is first bound to the receptor. A competitive analyte is then added to the sensing ensemble leading to recovery AO fluorescence via indicator displacement. Therefore, TNP can be successfully determined by a competitive fluorescence method based on a host–guest competitive recognition. The competitive fluorescence sensing platform based on PCP6-rGO is illustrated in Figure 1. This method is simple, low cost, sensitive, selective and has been successfully applied to TNP detection in tap water, lake water and soil samples.

## 2. Experimental Section

### 2.1. Reagents

Graphene oxide (GO) was obtained from Nanjing XFNANO Materials Tech Co., Ltd. (Nanjing, China). Acridine orange (AO) and TNP were acquired from Sigma Chemical Co. (St. Louis, MO, USA). PCP6 was synthesized according to the literature [30,31], and the synthetic route is shown in Appendix A. The structure and purity of all compounds were confirmed by ^1^H NMR and ^13^C NMR (see Appendix A). Other chemicals are of analytical grade. Deionized water (DW, 18 MΩ cm) was used to prepare all of the aqueous solutions.

### 2.2. Apparatus and Instruments

The samples are characterized by Fourier transform infrared (FTIR) spectroscopy via the SCIENTIFIC Nicolet IS10 (Thermo Fisher Scientific, New York, USA) FTIR impact 410 spectrophotometer using KBr pellets at a wavelength of 4000–400 cm^−1^. Thermogravimetric analysis (TGA) at 25 to 800 °C with a heating rate of 10 °C min^−1^ in nitrogen is performed in the Q50 TGA (TA Instruments, New Castle, DE, USA). The X-ray photoelectron spectroscopy (XPS) is performed on an ESCALAB-MKII spectrometer (VG Co., London, UK) with Al Ka X-ray radiation as the X-ray source for excitation. The zeta potential of the sample is measured with a Malvern Zetasizer Nano series. Fluorescent titrimetric experiments are performed on a Hitachi F-4500 spectrophotometer (Tokyo, Japan).

### 2.3. Synthesis of the PCP6-rGO Hybrid Nanomaterial

The PCP6-rGO composite was synthesized by heating and stirring the GO suspension under strong alkaline conditions in the presence of sodium citrate and PCP6 [32,33]. GO (20 mg) was dispersed in a solution of PCP6 (20 mg) and sodium citrate (100 mg) in DW (50 mL) by sonication, and the mixture pH was adjusted to 12 by NaOH solution (1 M) and stirred at 90 °C for 5 h. The black dispersion of PCP6-rGO was separated by centrifuging with 13,000 rpm for 1 h and washed with DW three times to obtain PCP6-rGO composite. A control experiment for the prepared rGO was developed in absence of PCP6 solution, and the PCP6-rGO was stable for over six months. Photographs of PCP6-rGO and rGO aqueous dispersion are shown in Appendix A.

### 2.4. Fluorescent Experiments

Aqueous solutions of AO (100 μM), TNP (200 μM) and PCP6-rGO (1.0 mg mL^−1^) were prepared by DW, respectively. A final concentration of 10 μM AO was also obtained via dilution. PCP6-rGO was gradually added to the AO solution, and the fluorescence of the AO was gradually quenched. The competitive displacement experiments were as follows: the TNP solution was gradually added into a complex of AO@bound PCP6-rGO to displace the AO molecule from the cavity of PCP6. The fluorescence signal was measured and recorded after the combined solution was vortex mixed for 3 min.

## 3. Results and Discussion

### 3.1. Characterization of the PCP6-rGO Hybrid Nanomaterial

The GO and PCP6-rGO are characterized by UV-vis spectroscopy (Figure 2). The UV-vis spectroscopy of GO dispersion shows a strong adsorption at ~230 nm and the peak gradually red-shifts from 230 to 260 nm (PCP6-rGO), suggesting that GO was reduced to rGO by sodium citrate under strong alkaline conditions [34,35]. We further characterized the microstructure of PCP6-rGO by TEM (Figure 2B). The TEM image of rGO (Figure 2B) shows the aggregation by the π−π interaction between rGO and rGO. However, the TEM image of PCP6-rGO (Figure 2C) reveals that the material monodisperses thin, wrinkled sheets, which is caused by the dispersion capability of the supramolecular pillar[6]arene. It is difficult to distinguish the PCP6 on the surface of rGO, therefore, the PCP6-rGO is further characterized by FTIR, zeta, Thermogravimetric Analysis (TGA) and X-ray-photoelectron spectroscopy (XPS). The FTIR spectrums of rGO, PCP6-rGO and PCP6 are shown in Figure 3A. Firstly, the FTIR of rGO shows some weak adsorptions due to a small amount of the remaining oxygen-containing functional groups [27]. Secondly, the FTIR spectrum of PCP6-rGO and PCP6 are very similar. The bands located at 1590 cm^−1^, 1556 cm^−1^ and 1409 cm^−1^ are ascribed to the phenyl in PCP6. The bands at 1618 cm^−1^ and 1388 cm^−1^ are attributed to pyridine of PCP6. The FTIR results indicate that PCP6 has grafted on the surface of rGO to form PCP6-rGO composite. The PCP6-rGO is also characterized by the TGA (Figure 3B). The mass loss of rGO is only 6 wt % at 600 °C due to the decomposition of remaining oxygen-containing functional groups. The mass loss of the PCP6-rGO reaches approximately 56 wt % when the temperature is 550 °C. Thus, the mass loss caused by decomposition of PCP6 is to be 50.0 wt % when the mass loss of rGO is deducted. The zeta potential measurements of PCP6-rGO and rGO are obtained and demonstrated in Figure 3C. The zeta potential of PCP6-rGO is 35 mV, the increased zeta potential value is attributed to the large number of the positive charged pyridinium units in the PCP6 molecule. Moreover, the PCP6-rGO zeta potential is higher than 30 mV [36], indicating that the stability and dispersion of PCP6-rGO are very high.

The XPS results of GO and PCP6-rGO are provided in Figure 3D. A significant N 1s peak is observed for the PCP6-rGO sample, but no N signal is detected on the GO film, showing that rGO has been functionalized by PCP6. In addition, the intensities of C=C/C–C peaks in PCP6-rGO predominated upon comparing the C 1s of GO. Figure 3E,F shows that the intensities of all of the C 1s peaks of C–O, C=O and O–C=O obviously decrease indicating that the GO is reduced to rGO. Therefore, these results of FTIR, TGA, zeta potential and XPS suggest that PCP6 has successfully grafted on the surface of rGO by π−π interaction between rGO and PCP6 [37].

### 3.2. Fluorescence Spectra Analysis

We first study the fluorescence quenching ability of PCP6 and PCP6-rGO towards AO. While the fluorescence intensity of AO decreases with PCP6 via host-guest recognition interactions (Appendix A), the fluorescence intensity of AO quenching is mainly caused by PCP6-rGO based on the outstanding fluorescent quenching performance of rGO. The fluorescence quenching of the dye is caused by the fluorescence resonance energy transfer (FRET) between dye and graphene [32,33]. The host–guest recognition between PCP6 and AO was studied by UV-vis and shown in Appendix A. The AO exhibits an absorption peak at 490 nm, and the absorption intensity gradually decreases with the increasing of PCP6, and a new absorption is observed at 503 nm, and then a red shift occurs. These results are in accordance with the reported work by Hua and co-workers [38].

Figure 4A shows the fluorescence quenching performance with PCP6-rGO towards AO. The fluorescence of AO is continuously quenched with the increasing of PCP6-rGO. Figure 4B shows the successive reversal of the AO fluorescence signal with repeating TNP dosing on the pre-formed AO@PCP6-rGO inclusion complex and the concentration of PCP6-rGO is 18 µg mL^−1^. The fluorescence signal reversion is caused by the addition of the amount of TNP, which suggested the successful detection of TNP via this fluorescence approach. Therefore, we conclude that the AO enters into the cavity of PCP6 and forms an inclusion complex with PCP6-rGO.

Appendix A shows that the fluorescence of AO is recovered by the TNP (20 μM), and the effects are similar at 25 °C, 35 °C, 45 °C and 50 °C, indicating that the temperature change does not affect the sensing performance. In addition, the AO is released from the cavity of PCP6 by the addition of TNP based on the competitive supramolecular recognition. This is a “switch-off-on” fluorescence process. The AO is also incubated with PCP6-rGO to form an AO@PCP6-rGO complex that is attached to the rGO. This is accompanied by indicator fluorescence ‘turn off’ due to fluorescence resonance energy transfer (FRET) [32,33]. Some control experiments are performed to confirm that the observed fluorescent intensity recovery is caused by the displacement of AO by TNP from the cavity of PCP6 host molecule. Figure 5A,B shows that the fluorescence quenching phenomenon obviously occurs between rGO and AO, the fluorescence reversion does not recover upon the addition of TNP. Therefore, the dye indicator AO first combines with PCP6-rGO and is then released from the cavity of PCP6 upon the addition of TNP. This forms a “fluorescent switch”.

Figure 4C presents the calibration curves for the quantitative determination of TNP, and the fluorescence ratio F/F_0_ is proportional to the concentration of TNP. The linear response ranges for TNP detection are 0.01−5.0 and 5.0−125.0 μM. The detection limit is 0.0035 μM (S/N = 3), and the corresponding regression equations of F/F_0_ = 0.26 C (µM) + 0.65 and F/F_0_ = 0.02 C (µM) + 1.84 are also obtained. This approach is compared with other methods for the determination of TNP (Appendix A). This competitive fluorescent method shows a wider linear range, lower detection limit and high selectivity versus previously reported approaches. Moreover, this method is very convenient and simple for the determination of TNP.

### 3.3. The Analysis of Host–Guest Recognition

The three nitro groups are strong electron-withdrawing groups, thus trinitrophenol is an electron-deficient molecule. It is easily captured by the electron-rich cavity of the pillararenes. The nitro moiety on TNP makes it quite negative, and the molecular size of TNP is suitable for the PCP6 cavity that is captured by the positively charged pillar[6]arene via the electrostatic interactions, which plays an important role in host-guest recognition [37]. Therefore, TNP molecule can be easily recognized by cationic and electron-rich PCP6—this leads to high selectivity for TNP detection.

### 3.4. Selectivity and Practical Sample Analysis

The interference study for the detection of TNP with AO-bound PCP6-rGO is measured with 100-fold concentrations of TNP analogues (4-NP, 3-NP, 2-NP, 4-aminophenol, 3-aminophenol, 2-aminophenol, NB, 1,4-DNB, 1,3-DNB and 1,2-DNB) and common interferents (NaCl and K_2_SO_4_). The chemical structures of analogues are shown in Appendix A. Figure 4D shows that the fluorescence intensity does not change when these interferences are added to AO@PCP6-rGO by comparing TNP in the presence of AO@PCP6-rGO. Figure 4E shows a significant fluorescence increase upon the addition of TNP. However, the addition of other competitive interferents do not cause significant fluorescence changes. In addition, Figure 4F illustrates that the fluorescence intensity of AO does not change when other competitive interferents are added to a mixture of AO@PCP6-rGO and TNP. The fluorescence intensity of AO is slightly changed by the addition of TNT in Figure 4E,F. Although the structures of TNP and TNT are highly similar, the higher recognition capability between PCP6 and TNP than TNT dominates the recognition object that PCP6 preferentially combines with TNP due to the phenolic hydroxyl group in TNP, which offering electrostatic interactions between cationic pillar[6]arene and phenolic hydroxyl group. There is no phenolic hydroxyl group in TNT, indicating that the recognition ability between PCP6 and TNT is feeble. Therefore, the fluorescence intensity of AO is slightly changed by the addition of TNT. This demonstrates that these interferents do not cause a false-positive signal.

To assess AO@PCP6-rGO in practical applications, TNP is detected with standard addition method in two water samples and a soil sample. The procedure of preparing practical samples is elaborated as follows: Soil (50 mg) is added to DW (100 mL) under sonication, the solution is filtrated by filter membrane (0.45 μM) for removing insoluble solid matter. The tap and lake water samples are collected at tap water and lake water. The practical TNP detection is carried out by standard addition method in prepared soil sample. The procedure of detection TNP in water samples is similar with soil sample. The recoveries are 98.5–101.4%, and the RSDs are 1.7–5.8% in Table 1. The accuracy and precision of this proposed approach are satisfactory, which indicates that this method can be applied for the detection of TNP in water and soil samples.

## 4. Conclusions

In conclusion, we describe a simple, convenient and selective “switch-off-on” fluorescent sensing platform using pyridine-functionalized water-soluble cationic pillar[6]arene PCP6 and dye acridine orange AO nanocomposite as the energy donor-acceptor. The outstanding host–guest recognition capability of PCP6 and excellent quenching performance of rGO make this fluorescent sensing system suitable for TNP detection in tap water, lake water and soil samples. This work demonstrates that the PCP6-rGO composite is a good energy acceptor for fluorescence sensing platforms with potential applications in many fields.

## Figures and Tables

**Figure 1 sensors-19-00091-f001:**
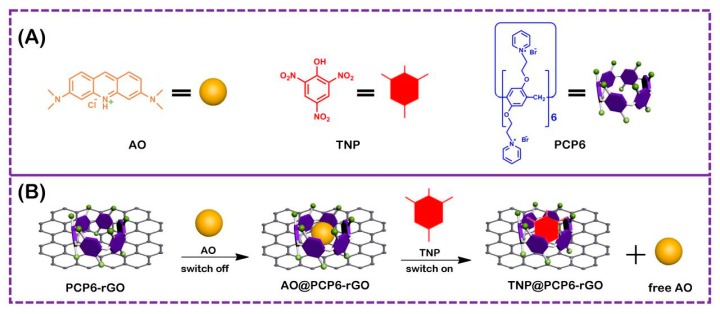
The corresponding cartoon representations of acridine orange (AO), trinitrophenol (TNP) and pillar[6]arene (PCP6) (**A**), and the illustration of PCP6-functionalized reduced graphene (PCP6-rGO) nanohybrids-based fluorescent sensing method towards trinitrophenol (**B**).

**Figure 2 sensors-19-00091-f002:**
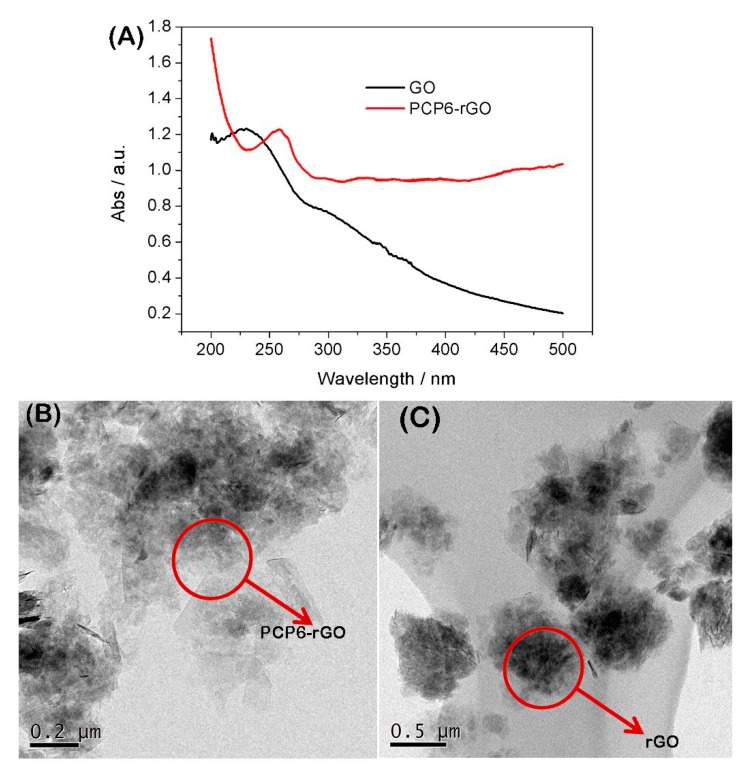
UV-vis spectra of the graphene oxide (GO) and PCP6-rGO in aqueous solution (**A**); TEM image of PCP6-rGO (**B**) and rGO (**C**).

**Figure 3 sensors-19-00091-f003:**
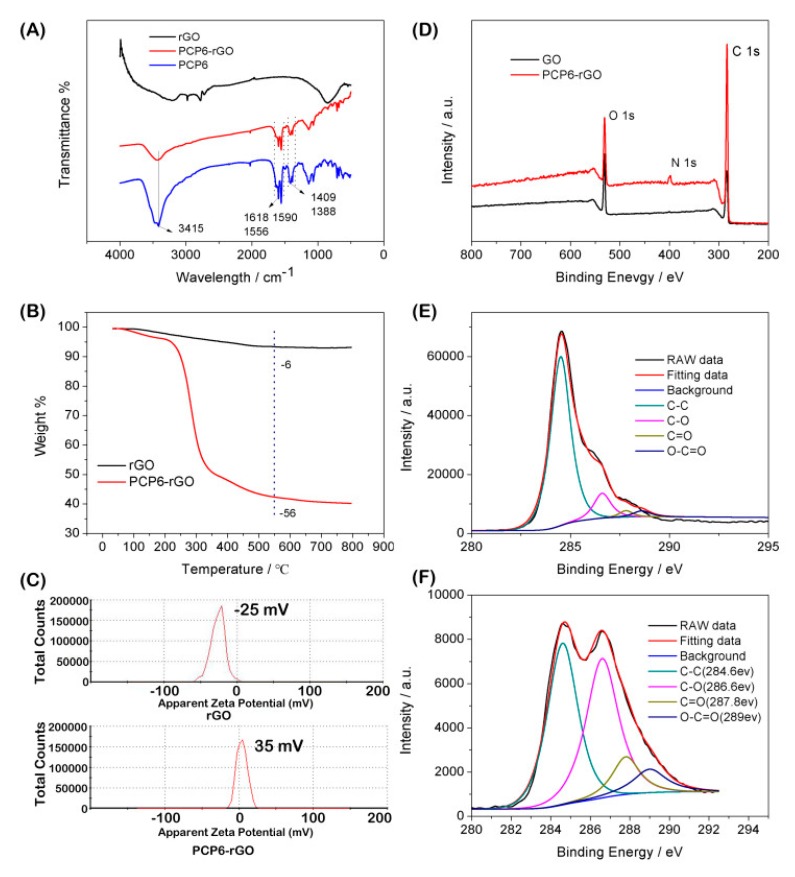
FTIR spectra of rGO, PCP6-rGO and PCP6 (**A**). Thermogravimetric Analysis (TGA) curves of rGO and PCP6-rGO (**B**). Zeta potentials of rGO and PCP6-rGO (**C**). X-ray-photoelectron spectroscop (XPS) survey spectra of GO and PCP6-rGO (**D**). The C1s XPS spectra of GO (**E**) and C1s XPS spectra of PCP6-rGO (**F**).

**Figure 4 sensors-19-00091-f004:**
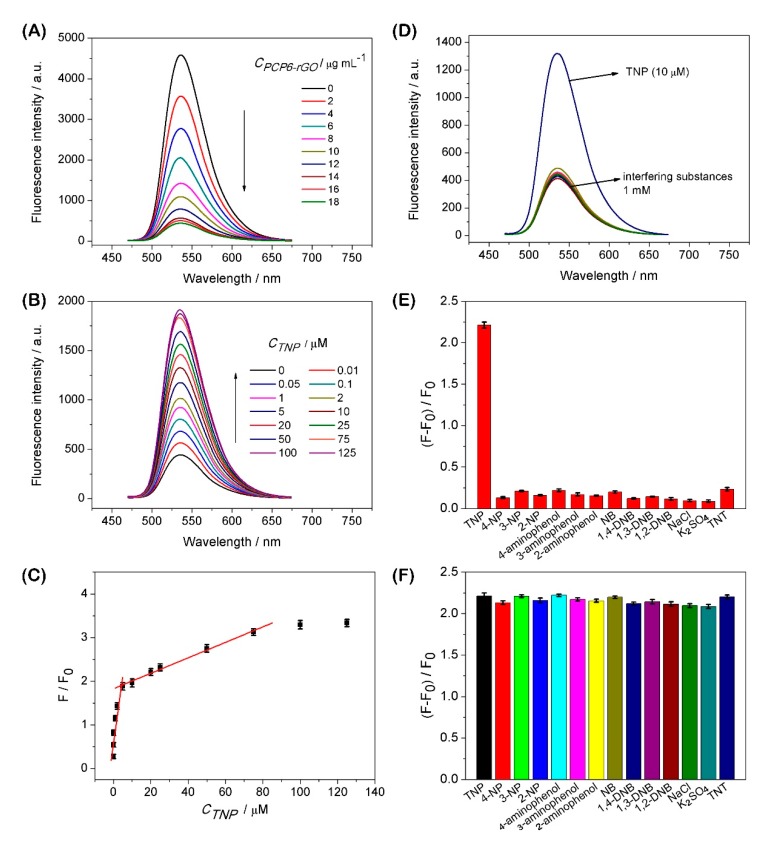
The effect of increasing concentrations of PCP6-rGO (concentrations ranging from 0 µg mL^−1^ to 18 µg mL^−1^) on the fluorescence intensity of AO (λex = 460 nm) dispersion (**A**). Fluorescence spectra of the AO@PCP6-rGO complex via the different concentrations of TNP and the concentration of PCP6-rGO is 18 µg mL^−1^ (**B**). Calibration curves of fluorescent intensity for AO@PCP6-rGO vs. TNP concentrations (**C**). Fluorescent spectra of AO@PCP6-rGO in the presence of TNP and others interferences and the concentration of PCP6-rGO is 18 µg mL^−1^ (**D**). The relative fluorescence intensity of (F − F_0_)/F_0_, and the F_0_ and F are the fluorescence intensity without and with the presence of 10 μM TNP and 1 mM interferences and the concentration of PCP6-rGO is 18 µg mL^−1^ (**E**). The relative fluorescence intensity [(F − F_0_)/F_0_] of 10 μM TNP in the absence 1 mM others interferences and the concentration of PCP6-rGO is 18 µg mL^−1^ (**F**).

**Figure 5 sensors-19-00091-f005:**
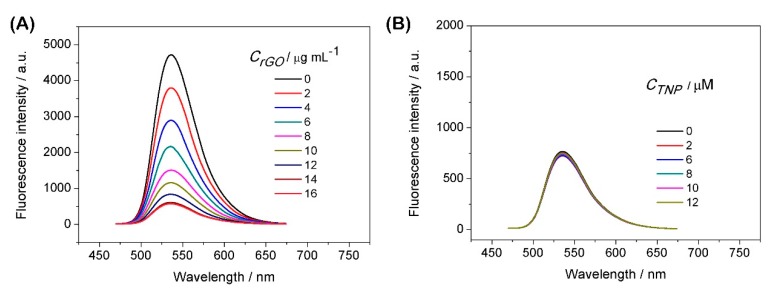
(**A**) The effect of increasing concentrations of rGO (concentrations ranging from 0 to 16 µg mL^−1^) on the fluorescence intensity of 10 µM AO (λex = 460 nm). (**B**) Fluorescence spectra of the AO@rGO complex via different concentrations of TNP and the concentration of rGO is 16 µg mL^−1^.

**Table 1 sensors-19-00091-t001:** Determination of trinitrophenol (TNP) in tap water, lake water and soil samples (n = 3).

Sample	Added (μM)	Found (µM)	RSD (%)	Recovery (%)
Tap water	0	0	-	-
	2	1.95 ± 0.06	3.1	97.5
	5	5.07 ± 0.15	2.9	101.4
	10	9.89 ± 0.32	3.2	98.9
Lake water	0	0	-	-
	2	1.97 ± 0.11	5.5	98.5
	5	4.97 ± 0.09	1.8	99.4
	10	10.09 ± 0.21	2.1	100.9
Soil	0	0	-	-
	2	2.06 ± 0.12	5.8	103
	5	4.97 ± 0.19	3.8	99.4
	10	9.95 ± 0.17	1.7	99.5

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
