# Peer review of "A Fluorescence Sensing Determination of 2,4,6-Trinitrophenol Based on Cationic Water-Soluble Pillar[6]arene Graphene Nanocomposite"

_sensors, 2018, doi:10.3390/s19010091_

Round 1
Reviewer 1 Report
In this manuscript, the authors demonstrated the use of functionalized reduced graphene oxide (rGO) as fluorescence receptors for detecting trinitrophenol (TNP) molecules. The surface of rGO is conjugated with pyridine-functionalized pillar[6]arene (PCP6). Due to the complementary binding of probe molecules (acridine orange, AO) and PCP6 on the rGO, the fluorescence of AO is quenched. By introducing the TNP molecules, the fluorescence intensity is recovered. Thus, a linear signal response of 0.01−5.0 uM and a low detection limit of 0.0035 μM can be achieved. This work looks interesting and promising. I recommend the paper for publishing in this journal after the below concerns have been addressed:
Comments:
1. In Page 2, the authors have indicated that they have chosen the indicator/dye molecule as AO bound to the receptor. However, they did not explain the reason why they choose the AO molecules and will the detection ability be same with other dye molecules?
2. In Page 3, Figure 2. “UV-vis spectra of the GO and PCP6-rGO in aqueous solution (A); TEM image of PCP6-rGO.” The authors did not mention the size and diameter of the rGOs before and after the surface functionalization. A more clear TEM or SEM figure should be provided to illustrate the GO dimension with different scale bar. And what is the size distribution and uniformity?
3. In Page 6, Figure 4. “The effect of increasing concentrations of PCP6-rGO (concentrations ranging from 0 μg/mL to 18 μg/mL) on the fluorescence intensity of AO (λex = 460 nm) dispersion (A). Fluorescence spectra of the AO@PCP6-rGO complexes via different concentrations of TNP (B). Calibration curves of fluorescent intensity for AO@PCP6-rGO vs. TNP concentrations (C). Fluorescent spectra of AO@PCP6-rGO in the presence of TNP and others interferences (D). The relative fluorescence intensity of (F-F0)/F0, and the F0 and F are the fluorescence intensity without and with the presence of 10 μM TNP and 1mM interferences (E). The relative fluorescence intensity [(F-F0)/F0] of 10 μM TNP in the absence 1mM others interferences (F).” The authors have compared the detection of the TNPs with other substances to show the selectivity of the probes. Is it possible to compare with the similar species such as TNT molecules? Will the signal be differentiate?
4. The authors miss several current papers in their citation "An unusual “off-on” fluorescence sensor for iron (III) detection based on fluorescein–reduced graphene oxide functionalized with polyethyleneimine." Sensors and Actuators B: Chemical 239, 343, 2017. "New generation cadmium-free quantum dots for biophotonics and nanomedicine" Chemical Reviews 116, 12234, 2016.
Author Response
Dear reviewer,
Thank you so much for giving us an opportunity to revise our manuscript (No: sensors-405234 entitled “ A fluorescence sensing determination of 2, 4, 6-trinitrophenol based on cationic water-soluble pillar[6]arene graphene nanocomposite”). We are grateful for the detailed comments and suggestions provided by each of the reviewers, and we believe that their input has greatly improved our revised manuscript. Changes were marked in yellow in the revised manuscript.
We would like to publish in Sensors. All co-authors have seen and agreed with the contents of the revised manuscript and there is no financial interest to report. We certify that the submission is not under review at any other publication.
Please let me know your decision at the earliest convenience.
Sincerely yours.
Xiao-ping Tan, PhD

Reviewer 2 Report
This manuscript demonstrates a fluorescence platform for the detection of trinitrophenol based on competitive host–guest recognition between pyridine-functionalized pillar [6]arene and acridine orange where PCP6-functionalized reduced graphene (PCP6-rGO) is used as a receptor. This platform was also used to test TNP in two water and soil samples. Authors claim that PCP6-rGO composite is a good energy acceptor for fluorescence sensing platforms ant their platform has a low detection limit and a wide linear response.
The topic is relevant, however, there are some issues which need to be addressed before publication.
1)p.6, Fig.4a shows how the fluorescence intensity of AO decreases with increasing concentration of PCP6-rGO, whereas Fig.4b shows what happens after adding TNT. It is not clear which concentration of PCP6-rGO was used to prepare Fig.4b.
(i) How the concentration of PCP6-rGO affects the trend presented in Fig.4b? How the fluorescence spectra of AO@PCP6-rGO changes with adding TNT for 2 \micro g mL^-1 and 18 \micro g mL^-1 of PCP6-rGO?
2)p. 7, Fig.5. Again, it is not clear which concentration of rGO was used to prepare Fig.5b.
3)p. 7, ' Fig.4C presents the calibration curves for the quantitative determination of TNP, and the 194 fluorescence ratio F/F0 is proportional to the concentration of TNP. The linear response ranges for 195 TNP detection are 0.01−5.0 and 5.0−125.0 μM.' Authors should comment more on the presented trend. In particular, they should explain the origin of three different trends visible in Fig.4c: below 10 \micro M of TNT, between 10 and 80 \micro M of TNT and above 80 \micro M of TNT.
4)p. 7/8, 'Selectivity and practical samples analysis', It is not clear what was the concentration of PCP6-rGO employed for those tests.
Therefore, I recommend this manuscript for publication in Sensors after a major review.
Author Response

(The authors gave the same response as above.)

Reviewer 3 Report
This work present the detection of 2, 4, 6-trinitrophenol using pillar[6]arene reduced graphene oxide as a fluorescence probe. This is an interesting work with novel materials and interesting application. However, before publication, it requires a major revision. A better description why this probe is better than previous works (include a table) is required. Why did you use reduced graphene oxide and not graphene oxide (since the latter is known to have better quenching capabilities)? Also, the manuscript only mentions the advantages but what about the challenges of the work and possible ways for improvement (more discussion is required in this sense, example, this is not a sensor since the probe is not reversible, not a continuous switch).
More specific comments:
- Title: Remove the words “facile, high selectivity and novel”, the paper should be novel to be published in the journal. Analytical features should not be key words in the title instead, method, analyte.
- Abstract: Include the units for the TNP calibration range.
- Introduction: In the statement, “Unfortunately, these methods often need a complicated synthetic process or labeling procedure—these are time consuming, expensive and complicated.” I don’t agree that electrochemical methods require labeling (TNP is electroactive in carbon for example), or are expensive.
- Introduction: Some of the statements should be tone down, “none are suitably fast, sensitive and selective”.
- There are some errors concerning the use of sensing. Since, this probe seems to be an irreversible sensor (no data is shown to be reversible), it should be denoted as a fluorescence probe rather than a sensor, because it does not allow reversibility nor monitoring.
- Figure 3. Panel A, include the concentration of AO used in this analysis. Panel C is blurred, not clear what it is shown in axis X and Y. Panel D and E, are you representing GO or rGO, in the caption says GO, but in the text rGO. Caption of panel F is unclear.
- Table 1, include the number of samples tested.
- English should be improved.
Author Response

(The authors gave the same response as above.)

Round 2
Reviewer 1 Report
The manuscript has been improved after revisions according to the reviewers' comments. I recommend the paper to be published in the present form.
Author Response
Dear reviewer,
Thank you for your approval of our manuscript.

Reviewer 2 Report
I am completely satisfied with the supplied corrections. I recommend this paper for publication in Sensors in the present form.
Author Response

(The authors gave the same response as above.)

Reviewer 3 Report
Although authors have made a good effort to accomplish the majority of the concerns. There are still few points that should be revised before publication.
The most important is that I consider that authors have to change the Figure 4E and include TNT, along with necessary discussion in the manuscript, this is a very relevant point to increase the interest of future readers.
Few more minor points:
1) I consider that the TEM image showed includes at the right bottom structures mainly based on rGO and the darken needles on the left are mostly PCP6-rGO. Discussion and or TEM images with separate material will enhance the knowledge of the new hybrid nanomaterial.
2) Modify Figure 3C, so the plot is not blurred.
3) In page 5, replace occurrs by occurs.
4) Limit the number of references to around 30-40. Only relevant references should be included. If not move some of the literature review to ESI.
Author Response
Dear reviewer, Thank you so much for giving us an opportunity to revise our manuscript (No: sensors-405234 entitled “ A fluorescence sensing determination of 2, 4, 6-trinitrophenol based on cationic water-soluble pillar[6]arene graphene nanocomposite”). We believe that your input has greatly improved our revised manuscript. Changes were marked in yellow in the revised manuscript. We would like to publish in Sensors. All co-authors have seen and agreed with the contents of the revised manuscript and there is no financial interest to report. We certify that the submission is not under review at any other publication. Please let me know your decision at the earliest convenience. Sincerely yours. Xiao-ping Tan, PhD

Round 3
Reviewer 3 Report
Authors have highly improved the manuscript and now it can be consider for publication.